# Psychological Changes in Green Food Consumption in the Digital Context: Exploring the Role of Green Online Interactions from a Comprehensive Perspective

**DOI:** 10.3390/foods13183001

**Published:** 2024-09-22

**Authors:** Siyuan Zhang, Shiwei Xu, Yilei Ren, Jing Wang

**Affiliations:** 1College of Economics and Management, Shanghai Ocean University, Shanghai 201306, China; m220751374@st.shou.edu.cn (S.Z.); swxu@shou.edu.cn (S.X.); 2School of Economics and Management, Tongji Zhejiang College, Jiaxing 314051, China; 3SinoDe Consulting eK, 60327 Frankfurt, Germany

**Keywords:** green food, consumer choices, online green interaction, psychological mechanisms

## Abstract

The advent of the digital economy has brought new opportunities to food marketing. In China, many food businesses have begun to use interactions under specific social media topics to open up new sales channels. Green food, as a representative of environmentally related topics, is increasingly influencing consumer choices through online interactions. In light of this, this study collected data from a large group of participants engaged in online green interactions to explore the psychological mechanisms behind consumers’ choices of green food in an online context. The findings indicate that online green interactions positively influence the willingness to purchase green food, with environmental self-efficacy and flow experience serving as mediators in this relationship. Information trust and consumer traits act as boundary conditions. This study not only deepens the understanding of food consumer behavior in the digital context, but also provides important references for food companies on how to more effectively utilize online interaction to promote the market expansion of green food.

## 1. Introduction

Since ancient China, the adage that the cornerstone of life resides in food has persisted, imbuing the realm of food with profound cultural significance. Unfortunately, global health statistics painted by the World Health Organization (WHO) underscore a grim reality: annually, 600 million people succumb to foodborne illnesses, underscoring the urgent need for food safety [1]. Ensuring the safety of our plates is not only vital for public health but also pivotal in achieving food security. 

In the early 1970s, the “organic agriculture” movement, which began in the United States and spread to countries in Europe and Asia, had a significant impact on many nations [2]. The goal of this movement was to curb the excessive use of chemicals to protect the environment and enhance food safety. In response, several countries began to support their domestic enterprises in developing and producing pollution-free food through economic and legal measures [3]. In China, “green food” emerged as a representative product influenced by this movement. Green food refers to safe, high-quality, edible agricultural products and related items produced in a favorable ecological environment, adhering to green food standards and undergoing comprehensive quality control throughout the production process [4,5]. This ensures that these products are granted the right to use the green food logo [6]. In recent years, China’s green food industry has seen steady and rapid growth. According to data from the China Green Food Development Center (http://www.greenfood.agri.cn), by the end of 2022, the number of registered entities effectively using the green food logo had reached 26,000. However, despite this progress, the domestic consumption of green food remains below 1% of total food intake, significantly lower than the global average of 2% [7]. Recognizing this disparity, the National Development and Reform Commission has charted a course with the “Implementation Plan for Promoting Green Consumption”, aimed at fostering a healthier and more sustainable food consumption mindset among the populace. Therefore, encouraging consumers to purchase green food is a central focus for government efforts to promote a shift towards sustainable consumption and a significant challenge for enterprises aiming to excel in green food marketing and drive strategic transformation toward sustainable practices. Therefore, investigating consumers’ willingness to purchase green food is of great practical significance in unlocking and stimulating the substantial potential demand for green food among consumers.

In the process of shaping human behavior, individual humans can exhibit subjective initiative and free will, that is, the ability to control and become the subject of behavior [8]. In addition, Individual behavioral intentions are context-dependent and inevitably shaped by external factors [9]. So, human behavior is shaped by the complex interplay between environmental factors and individual free will. Media promotional activities can influence consumer behaviors related to purchasing environmentally friendly products, recycling waste, and other pro-environmental actions [10,11]. Thus, as a key channel for information dissemination, the media holds significant potential to impact green food consumption behavior. With the advent of the internet and the rapid growth of social media, the ways people communicate and interact have expanded beyond traditional temporal and spatial constraints [12]. According to the Statistical Report on the Development of China’s internet, by 2023, the number of internet users in China reached 1.079 billion, with a penetration rate of 76.4%. The evolution of social media and internet technologies has significantly increased online interactivity. This new form of social interaction has garnered particular attention in marketing and sociology. The exchange of information within these social networks has had a profound influence on consumer behavior [13]. 

Recent academic research has increasingly focused on the impact of online interaction on consumer behavior [14]. However, most studies concentrate on general consumption patterns and seldom address the specific domain of green food consumption. Additionally, existing research primarily explores interactions between businesses and consumers, with less emphasis on consumer-to-consumer interactions. Consumers now seek more than just the product details provided by businesses; they desire a deeper understanding through online reviews and daily exchanges, which enhances their shopping experience. Moreover, the mechanisms of interaction between businesses and consumers may differ from those among individual consumers [15]. In the literature on green food consumption, most analyses have approached consumer behavior from a rational cognitive perspective. For example, Konuk interprets the behavior of Türkiye consumers to buy organic food by perceiving the rationality of food quality and price [16]. Bulsara believes that environmental concerns are an important reason that Indian consumers choose organic food [17]. From the above perspective, most research is conducted through the understanding of products or environments. Emotional factors have not been included in the scope of examination abroad, and even most Chinese scholars have overlooked this point. However, Eastern collective cultures, which prioritize emotions and social harmony more than Western material cultures, play a significant role in the decision-making process of Chinese consumers. Thus, the affective decision-making perspective is crucial and should not be overlooked by researchers [18]. Additionally, it is important to explore how user personality traits and the degree of information trust influence these transmission mechanisms.

Interactivity has positively influenced consumer purchasing behavior, enhanced product quality and creativity, and contributed to accelerating economic growth [19]. However, there remain significant research potential and opportunities for further development in leveraging new media marketing to boost consumer willingness for green purchasing and promote widespread green consumption. This study, therefore, utilizes the “Stimulus-Organism-Response” (SOR) theoretical model as its foundational framework, integrating social cognitive theory (SCT) and flow theory. It examines the impact of online green interactions on consumers’ willingness to purchase green food, adopting a composite perspective of “emotion-cognition-consumer personality”.

Based on the above analysis and discussion, this article aims to address the following two questions: (1) Can online green interaction serve as a new incentive channel to promote consumers’ purchase of green food? (2) How does consumer psychology change during this process?

The subsequent structure of this article is outlined as follows: Section 2 comprises theoretical analysis and the formulation of research hypotheses. Section 3 delves into the research design. Section 4 presents the empirical analysis conducted. Lastly, Section 5 encapsulates the research conclusions and discussions.

## 2. Theoretical Basis and Research Hypotheses

This study is grounded in the “Stimulus-Organism-Response” (SOR) theoretical model. Originating from the “Stimulus-Response” behavior theory proposed by Mehrabian and Russell, the SOR model is a foundational concept in modern cognitive psychology. The SOR theory posits that organisms mediate the relationship between stimuli and responses; external stimuli induce changes in an individual’s psychological state, which in turn lead to corresponding behavioral responses [20]. In this model, “S” represents external stimuli, referring to individuals’ perceptions of their environment and serving as critical inputs for decision making. “O” denotes internal psychological states, encompassing perceptions, emotions, and cognitive processes. “R” is the behavioral outcome, reflecting the actions consumers take in response to external environmental changes and their psychological reactions.

A review of research utilizing the SOR framework reveals its strong applicability in studying consumer behavior, particularly in online contexts. Therefore, this study adopts the SOR framework to analyze how online interactions influence consumers’ willingness to purchase by triggering specific psychological responses.

### 2.1. Online Green Interaction (OnlGreInt) and Green Food Consumption Intention (GreFConInt)

Interactivity, or social interaction, refers to the dynamic process of maintaining mutual interdependence among various social components, such as individuals or groups [21]. As one of the most fundamental aspects of human practice, interaction has been integral to human production and life throughout history. With the advent of the digital age and the rise of social media, interaction has acquired a new dimension—online interaction, which refers to communication mediated by the internet [22]. This shift to online interaction has transformed interpersonal communication by freeing it from the limitations of physical proximity. For consumers, online interaction has been pivotal in moving the exchange of product information from offline settings to online interactive platforms [23].

As online interaction becomes increasingly ingrained in modern life, it inevitably influences individual consumer decision making and behavior. From the perspective of online interactions between businesses and consumers, Jiang argues that in B2C online shopping, online interaction can shape customers’ impulsive buying behavior [24]. Similarly, Roggeveen et al. suggest that merchants can enhance consumer engagement and strengthen purchase intentions by using visually rich methods to showcase products on online platforms [14]. From the perspective of online interactions between individual consumers, Wu and Keysar contend that such interactions can positively impact customer satisfaction [25]. Adjei et al.’s study further shows that the exchange of interactive information among users can effectively stimulate purchasing behavior [15]. In the context of green consumption, Sheng et al. suggest that post-purchase feedback interactions allow businesses to effectively communicate value information to consumers, thereby encouraging repeat engagement in green consumption [26]. Additionally, Wang et al. demonstrate that both online informational and affective interactions among users significantly influence green consumption behavior [27].

The research outlined above suggests that both interactions between businesses and consumers, as well as interactions among individual consumers, significantly influence purchase decisions. In the context of green food consumption, online interactions act as external stimuli within the SOR model, driving the behavioral factors that shape consumers’ willingness to purchase green food products. Based on this, this paper proposes Hypothesis 1:

**H1:** *Consumer online green interactions positively influence the willingness to purchase green food*.

### 2.2. The Mediating Role of Environmental Self-Efficacy (EnvSelEf)

Self-efficacy refers to an individual’s judgment, belief, or subjective confidence in their ability to successfully accomplish a specific task at a certain level [28]. Environmental self-efficacy is an extension of this concept within the context of environmental conservation. Robert defines environmental self-efficacy as an individual’s perception and assessment of their ability to address environmental and resource-related challenges [29], specifically their cognitive perception of their capacity to engage in environmentally friendly behaviors. According to social cognition theory, human behavior reflects an individual’s cognition of themselves and their environment [30]. Studies have shown that environmental self-efficacy positively affects the intention to engage in green purchasing [31]. Research has also demonstrated that environmental self-efficacy has greater predictive and explanatory power than environmental concern, highlighting that, in addition to environmental cognition, self-cognition significantly influences behavior. Individuals with high self-efficacy are more likely to adopt pro-environmental behaviors when addressing problems or completing tasks [32]. The factors influencing self-efficacy have been thoroughly discussed in the academic community. Bandura’s research indicates that, beyond direct experiences, indirect vicarious experiences and social persuasion can also impact an individual’s self-efficacy [33]. Self-efficacy serves as an intermediary between knowledge, skills, experience, and subsequent behavior; the knowledge, skills, and experience an individual possesses directly influence their self-efficacy [34].

Media also plays a role in shaping self-efficacy, as the information it disseminates can increase individuals’ awareness of environmental issues. The more green product information is shared, the more likely individuals are to gain confidence in purchasing green products. When users engage in social media interactions regarding green food, they can acquire knowledge, skills, and experiences shared by others. Additionally, individuals who participate in environmental actions or prefer green food can encourage others to join in these interactions and activities [35]. As individuals gain more knowledge, deeper understanding, and experience with green food, and as they are influenced and encouraged by others, their unfamiliarity with and rejection of the concept decreases. Consequently, they become more confident in approaching and engaging with green food, thereby enhancing their environmental self-efficacy. This, in turn, strengthens their willingness to purchase green food [36]. Furthermore, consumer interactions and exposure to information stimuli on social media can enhance the imitation of others’ green food consumption behavior, allowing individuals to recognize their own capability to engage in green food practices by emulating successful examples set by others [37].

Based on the analysis above, we propose that green efficacy can function as a key psychological variable within the rational cognitive framework of purchase decision-making mechanisms. Accordingly, this paper proposes Hypothesis 2:

**H2:** *Environmental self-efficacy plays a mediating role in the influence of consumer online green interactions on the intention to purchase green food*.

### 2.3. The Mediating Role of Flow Experience (FlowExp)

The proposal of flow theory has significantly contributed to the growth of positive psychology. Flow refers to a positive psychological state marked by challenges, intrinsic rewards, and pleasure. This state is characterized by heightened concentration, a sense of enjoyment, increased curiosity, and a distorted perception of time, where time seems to fly by [38]. According to Pace, flow represents a unique state of consciousness in which individuals become deeply immersed in enjoyable activities [39]. Interaction theory further posits that flow emerges from the dynamic relationship between individuals and their environment, and that it is associated with positive emotions [40]. Additionally, in the context of green consumption, scholars argue that emotional factors often exert a more direct and significant influence on environmentally friendly behavior [41]. 

Hoffman and Novak were pioneers in applying the psychological concept of flow experience to marketing, specifically in the study of online influence. They demonstrated that the interactive nature of online environments can facilitate a flow experience for consumers [42]. Since then, the concept of flow has been extensively used to explore practical issues within the context of the Internet. For instance, Koufaris’ research found that the intensity of online consumer interactions positively influences the level of consumer concentration [43]. Similarly, Hsu et al. identified a significant relationship between various aspects of website quality—such as information quality, system quality, and service quality—and users’ flow experience. Their findings suggest that flow experience can mediate the relationship between website quality and both customer satisfaction and purchase intentions [44]. Dong’s empirical study further supports this by showing that “Seeding & Guerrilla Marketing” on the internet can induce a flow state in consumers, thereby enhancing their purchase intentions [45]. Consequently, flow experience has become a valuable framework for understanding consumer behavior in the digital age. Additionally, based on the SOR (Stimulus-Organism-Response) theory, user interactions on social media can elicit pleasure, leading to a state of immersion that ultimately drives purchase behavior.

The literature review highlights that flow is a state of pleasurable absorption, which can unconsciously influence consumers’ decision making and behavior. This emotional state, marked by happiness and pleasure, can therefore be a crucial psychological variable in exploring how affective states impact consumers’ purchase decisions. Based on this, the paper proposes Hypothesis 3:

**H3:** *The online green interactive experience mediates the influence of flow experience on the intention to purchase green food*.

### 2.4. The Chain Mediation Effect

Individual cognition is inherently limited, making it challenging to make fully rational decisions. Consumer decisions often result from the interplay between cognition and emotions, and there may be underlying connections between these two aspects of an individual’s decision making process [46]. According to social cognitive theory, emotions form the foundation of social cognition and significantly influence individual cognitive processes [30]. Self-efficacy, which arises from individuals’ perceptions of themselves, is also shaped by emotional factors. Positive emotions, such as happiness and interest, tend to enhance self-efficacy, whereas negative emotions, like anxiety and tension, can reduce it [47]. Research by Wu et al. suggests that trait flow positively impacts self-efficacy [48]. Similarly, Sheng et al. found that positive emotions can interact with individual environmental self-efficacy, thereby amplifying the effect of self-efficacy on green purchasing intentions [49]. 

This study suggests that emotions play a significant role in shaping self-efficacy within the context of food consumption. When consumers are in a positive emotional state, they are more confident in their ability to contribute to environmental protection through green food consumption and are more inclined to engage in green purchasing behavior. In contrast, consumers experiencing negative emotions are more likely to feel powerless regarding environmental issues and may doubt their capacity to make a meaningful impact, which can hinder the formation of green purchasing intentions.

Building on this analysis, the study posits a potential chain reaction in the process leading from interaction to the willingness to consume green food. Specifically, individuals who engage in green interactions may enter a state of flow, which in turn fosters positive emotions. This positive emotional state can enhance individuals’ sense of self-efficacy, thereby increasing their motivation to engage in green food consumption. Based on this reasoning, the study proposes Hypothesis 4:

**H4:** *Self-efficacy and flow experience play a mediating role in the relationship between online green interaction and willingness to consume green foods*.

### 2.5. The Moderating Role of Consumer Innovativeness (ConInn)

Trait theory suggests that individual behavior tendencies are shaped by personal traits, with individual innovativeness playing a crucial role in influencing consumer decision-making behavior. Innovativeness, as a consumer trait, refers to a heightened sensitivity to technological advancements and a strong desire to purchase new products. This trait often leads consumers to become early adopters of innovative products [50]. 

In the field of marketing, numerous scholars have established the correlation between consumer innovativeness and emerging consumer behaviors. Consumer innovativeness has been shown to significantly influence the purchase of novel products [51]. For instance, Bartels and Reinders explored the relationship between consumer innovativeness and social roles in the context of green organic food consumption [52]. Lao examined consumer innovativeness as a precursor variable, discussing its relationship with green consumer behavior within the framework of the Theory of Planned Behavior (TPB) [53]. Empirical research has predominantly considered consumer innovativeness as an antecedent variable. However, in recent years, scholars have begun to explore its role as a moderating variable in consumer behavior. For example, Hur’s study found that consumer innovativeness positively moderates the effect of emotional value on purchase intention [54]. Additionally, other studies have identified a moderating effect of consumer innovativeness between cognition and behavior [55,56]. In light of these findings, this paper introduces consumer innovativeness as a moderating variable within the model to examine its impact on green food purchasing behavior.

Consumer innovativeness can significantly influence green consumption behavior [57]. Green products not only emphasize resource conservation and environmental protection but also often incorporate advanced technologies, leading to novel functional features and innovative design esthetics. In some cases, these products can even set new trends within their respective categories [58]. Both theoretically and practically, the innovativeness of individual characteristics is likely to impact consumers’ decisions to purchase green food. For consumers with a high degree of innovativeness, being early adopters and frequent consumers of green food serves as a way to express their individuality. Therefore, driven by this personality trait, the amplification of green interaction, flow experience, and self-efficacy can enhance purchase intention.

In this context, the paper explores the mechanisms through which rational cognition and emotions influence green food purchasing behavior. It introduces the consumer personality trait of innovativeness to examine its interactive effect on these processes. The goal is to broaden the research perspective to encompass the “cognition-emotion-consumer personality” framework. Based on this, the paper proposes Hypotheses 5–7. 

**H5:** *Consumer innovativeness plays a positive moderating role in the relationship between flow experience and intention to purchase green food*.

**H6:** *Consumer innovativeness plays a positive moderating role in the relationship between environmental self-efficacy and willingness to purchase green food*.

**H7:** *Consumer innovativeness plays a positive moderating role in the relationship between green interactivity and consumers’ willingness to purchase green food*.

### 2.6. The Regulatory Role of Information Trust (InfCre)

McKnight et al. argue that trust plays a crucial role in shaping individuals’ behavioral intentions [59]. In the marketing field, Chatterjee’s research further demonstrates that consumer trust significantly influences behavioral intentions [60]. With the advent of the internet era, the proliferation of food information has become increasingly complex, making it more challenging to distinguish between true and false information. This shift has resulted in a new focus in the food consumption market: the transition from consumers’ desire for information to their trust in that information [61]. 

Information trust refers to the subjective perception of recipients regarding the credibility of information disseminators throughout the entire process of information dissemination. It includes judgments about the personal traits of information sources and assessments of their credibility by the audience [62]. Consumers’ focus on the quality of online information, the quality of online interactions, and the credibility of information sources significantly impacts their consumption decisions [63].

Research by Fan and Wang indicates that the quality of online comment information positively moderates the influence of personalized intelligent recommendations on consumers’ online impulse buying intentions [64]. Conversely, low levels of consumer trust can impede purchasing decisions [65]. Liu et al. also argue that customers’ trust in information sources significantly influences their confidence in consuming traceable pork [61]. This body of research underscores the importance of trust in the digital information environment, especially in the context of online food consumption.

Based on the above analysis, this study suggests that when consumers have doubts about the authenticity of online green information, their acceptance of such information decreases, which in turn hinders the internalization of this knowledge and experience into self-efficacy. This skepticism may also lead to avoidance behaviors, such as refraining from making purchases. Conversely, when consumers trust green information, they are more likely to incorporate it into their own cognition, facilitating the rapid transformation of this information into personal efficacy. Accordingly, this study proposes Hypotheses 8–10:

**H8:** *Information trust plays a positive regulatory role in the relationship between flow experience and environmental self-efficacy*.

**H9:** *Information trust plays a positive regulatory role in the relationship between flow experience and willingness to purchase green food*.

**H10:** *Information trust plays a positive regulatory role in the relationship between online interaction and willingness to purchase green food*.

Overall, the schematic diagram of the proposed research model is presented in Figure 1:

## 3. Research Design

### 3.1. Data Sources

Given that the focus of this study is the consumption of green food in the context of online interactions, our target population is internet users. We distributed survey questionnaires through online platforms in China, specifically targeting individuals who engage in discussions on green topics or environmentally friendly food on platforms such as Weibo and various forums. This approach ensures a more accurate and relevant sample of survey participants.

The questionnaire is structured into three parts. The first part serves as a screening section, where respondents are asked whether they have purchased green food after being introduced to the concept and viewing several product images with “green food labels.” Only those who respond “yes” proceed to the subsequent questions, while those who answer “no” are exited from the survey. Data from respondents who do not pass this screening are excluded to ensure that only relevant participants are included.

The second part gathers basic demographic information, including gender, age, level of education, monthly disposable income, occupation, and other pertinent details. The third part comprises seven scales, measuring aspects such as online green interaction, intention to purchase green food, flow experience, environmental self-efficacy, consumer innovativeness, and information trust. These scales are assessed using a 5-point Likert scale, where respondents indicate their level of agreement, ranging from “strongly disagree” to “strongly agree”.

A total of 458 questionnaires from Chinese consumers were collected for this survey. After excluding respondents who had not purchased green food, as well as incomplete or non-standardized responses, the final sample consisted of 407 valid questionnaires. Specifically, there were 18 individuals in the total sample who had never purchased or consumed green food, and 33 individuals whose questionnaire responses were missing, had inconsistent logic, or had too many questions with the same answer, et al. Therefore, the invalid samples were excluded. This yielded a valid response rate of 88.9%. [(458 − 18 − 33)/458 × 100% = 88.9%]. Figure 2 shows the research process of this article.

Our research sample is different from the overall population structure of China. Given that younger individuals are more familiar with new online social media and engage more frequently in communication and interaction, the survey participants in this study were primarily aged 19–30, representing 54.5% of the sample. For the group over 40 years old, their social media usage gradually decreases as they age, and they are also not the main participants in the interaction of “environmental protection and green topics”. From the table, it can be seen that the group participating in online green interaction is mostly concentrated in higher education institutions and graduate students, which reflects the high enthusiasm of the highly educated group for online green interaction. The basic characteristics of the remaining participants are detailed in Table 1.

### 3.2. Variable Selection

The key variables in this study include online green interactivity, intention to purchase green food, flow experience, environmental self-efficacy, consumer innovativeness, and information trust. To ensure the reliability and validity of the questionnaire, the measurement items were derived from established scales in relevant research. Furthermore, adjustments were made to the wording of certain items to better align with the characteristics of green food consumption and the research topic, while preserving the original intent. The specific details are presented in Table 2.

## 4. Results

### 4.1. Analysis of Data Validity and Reliability

The study analyzed the reliability of the main latent variables using SPSS 27.0 software, based on a sample of 407 valid questionnaires. As shown in Table 3, The mean values of all variables are above 2.5, indicating that consumers generally hold a positive attitude towards online interaction and green food purchases, which preliminarily supports the rationality of our research topic. In addition, the SD value is within a reasonable range, indicating a small degree of dispersion, and the sample selected in this article is suitable for quantitative analysis. The Cronbach’s Alpha coefficients for the study’s variables—online green interactivity (0.834), flow experience (0.904), green purchase intention (0.812), environmental self-efficacy (0.894), information trust (0.958), and consumer innovativeness (0.880)—all surpassed the esteemed threshold of 0.7 for high reliability, thereby confirming the excellent reliability of the selected variables. Furthermore, the Average Variance Extracted (AVE) for each latent variable exceeded 0.5, while the Composite Reliability (CR) also surpassed 0.7, underscoring the robust convergent validity of the scales employed for each latent variable.

Before conducting the hypothesis testing and regression analysis, confirmatory factor analysis was performed on the latent variables to assess their discriminant validity. As shown in Table 4, the fit indices for the six-factor model (χ^2^/df = 2.321, RMSEA = 0.057, GFI = 0.912, CFI = 0.969, IFI = 0.969, TLI = 0.962) met the criteria for model fit and were superior to those of other competing factor models. Therefore, the six variables selected for this study demonstrate strong discriminant validity.

### 4.2. Descriptive Statistics of Variables and Pearson Correlation Coefficient

After conducting a preliminary analysis of the distribution characteristics of each variable, this paper proceeds with a correlation analysis of the main variables to initially explore the relationships among them. The analysis results are shown in Figure 3. A notable and positive correlation has been established between online green interaction and consumer inclination towards purchasing green food, with a significant statistical value of r = 0.628 and *p* < 0.01. Furthermore, the majority of variables exhibit robust correlations among themselves, offering a preliminary affirmation of the research hypotheses formulated in this study. The Pearson correlation coefficient analysis outcomes solidly underpin the subsequent hypothesis testing phases. 

### 4.3. Regression Analysis

This section employs SPSS 27.0 to conduct regression tests on the research hypotheses proposed earlier. The results are presented in Table 5. Model (1) serves as the baseline model, which includes only four demographic variables: age, gender, income, and education level. The R-squared value is approximately 0.024. Model (2) adds the key independent variable, Online Green Interaction (OnlGreInt), to the baseline model. The results demonstrate a significant positive effect of OnlGreInt on the dependent variable (r = 0.812, *p* < 0.01). The overall model is significant, and the R-squared value significantly improves compared to the baseline model, thereby confirming H1.

We employed the PROCESS v4.1 plugin to examine the mediating effects proposed in Hypotheses 2–4. Model (3) tested the mediating role of flow experience (FlowExp) in the relationship between online green interaction and the intention to consume green food. The indirect effect coefficient for flow experience was 0.457, with a 95% confidence interval (CI) of [0.350, 0.580]. Since this interval does not include zero, it indicates that flow experience significantly mediates the positive relationship between online green interaction and the intention to consume green food, thereby supporting H2.

Model (4) examined the role of environmental self-efficacy (EnvSelEf) and found that it has a positive and significant direct effect on the intention to purchase green food (r = 0.466, *p* < 0.01). The effect of online green interaction (OnlGreInt) on the intention to consume green food remained significantly positive, though with a smaller coefficient compared to Model (2). Additionally, the indirect effect coefficient of environmental self-efficacy on the intention to purchase green food was 0.390, with a 95% CI of [0.291, 0.505], confirming H3.

Model (5) introduced both flow experience and environmental self-efficacy as mediators to investigate the presence of a chain mediation effect. The results showed a total indirect effect coefficient of 0.602, with a 95% CI of [0.480, 0.730]. Since this interval does not include zero, it confirms that flow experience and environmental self-efficacy mediate the relationship between online green interaction and the intention to consume green food in a chain-like manner, thus supporting H4.

The results of the moderation effect are presented in Table 6. According to the regression analysis results of Model (1) shown in the table below, the intention to consume green food is significantly influenced by the interaction between consumer innovativeness and flow experience (β = 0.118, *p* < 0.01). Furthermore, when comparing the mean values of consumer innovativeness at levels below and above one standard deviation, it is evident that as the value of the moderating variable consumer innovativeness increases, the effect of the independent variable on the dependent variable becomes more pronounced. Thus, it can be concluded that consumer innovativeness has an enhancing moderating effect on the relationship between consumer environmental self-efficacy and repurchase intention for green food, supporting H5. This finding demonstrates that consumer innovativeness plays a positive moderating role in the influence of flow experience on the intention to consume green food.

In Model (2), the interaction term between consumer innovativeness and environmental self-efficacy is also significantly positive. Similar to the findings in Model (1), when comparing the mean values of the moderating variable below and above one standard deviation, it is observed that as consumer innovativeness increases, the effect of the independent variable on the dependent variable becomes more pronounced, thus supporting H6.

Similarly, the examination results for H7, H8, H9, and H10 are represented by Models (3), (4), (5), and (6), respectively, and the data results support these hypotheses as well.

## 5. Discussion and Conclusions

### 5.1. Results and Comparison

This research analyzed 407 valid questionnaires using the SOR theory model, combined with social cognitive and flow theories, to investigate how online green interactions influence green food purchasing intentions. It examines the “emotion-cognition-consumer personality” framework to explore two key aspects:

Unveiling the psychological mediation process through both “rational cognition” and “emotion” pathways and identifying the chain-mediating effect between cognitive and emotional variables.

Introducing the moderating effects of consumer innovativeness and information trust to determine the boundary conditions for green food purchasing intentions. The specific research results are as follows.

Firstly, consumers’ online green interactions have a positive impact on the willingness to purchase green food, assuming Hypothesis 1 holds true. This conclusion is consistent with the research findings of Roggeveen et al. [14] and Adjei et al. [15]. They all believe that with the integration of online interaction into modern life, individual consumption decisions and behaviors are inevitably influence; whether it is the interaction between businesses and consumers or among consumers will affect consumer behavior to varying degrees.

Secondly, environmental self-efficacy and flow experience play a mediating role in the influence of consumer online green interactions on green food purchase intention, assuming Hypotheses 2 and 3 are valid. The studies conducted by Ge and Hu [35], as well as Bilgihan [40], respectively, confirm that consumer online interactions positively influence consumers’ environmental self-efficacy and flow experience. Additionally, the research conducted by Trivedi et al. [37] and Hsu et al. [44], respectively, demonstrates that environmental self-efficacy and flow experience lead to consumers’ purchase intention. These scholars merely elucidate the unidirectional relationship between consumers’ online green interactions, consumers’ willingness to purchase green food, and environmental self-efficacy without connecting them to analyze their relationship. Although some scholars have confirmed that the experience of flow plays a mediating role in the influence of consumers’ online green interactions on their willingness to purchase green food, the mediating variable is singular, failing to fully reflect the mechanism through which consumers’ online green interactions affect their willingness to purchase green food. This study confirms that consumers’ online green interactions not only have a direct impact on their willingness to purchase green food, but also demonstrate that the effect of consumers’ online green interactions on the willingness to purchase green food is achieved through the mediating role of environmental self-efficacy and the experience of flow.

Thirdly, environmental self-efficacy and flow experience play a chain-mediating role in the relationship between consumer online green interaction and green food consumption intention, assuming Hypothesis 4 holds true. Unlike previous studies that have examined environmental self-efficacy and flow experience as separate mediating variables, this finding confirms that individuals can achieve a state of flow and experience positive emotions when engaging in online green interactions. This positive emotional state, in turn, enhances individuals’ self-efficacy and triggers a chain reaction that promotes consumers’ willingness to consume green food. Moreover, this finding also supports the social cognitive theory, which suggests that emotions serve as the foundation of social cognition and have significant impacts on individuals’ cognition.

Fourth, consumer innovativeness plays a positive moderating role in the relationship among flow experience, environmental self-efficacy, online green interactivity, and willingness to purchase green food; that is, Hypotheses 5, 6, and 7 hold true. Information trust also plays a positive moderating role in the relationship between flow experience and environmental self-efficacy. Additionally, information trust moderates the relationships among flow experience, online interactivity, and willingness to purchase green food. It is assumed that Hypotheses 8, 9, and 10 hold true. The moderating role of consumer innovativeness and information trust is consistent with the findings of Hur et al. [54] and Calvo-Porral et al. [65]. From the perspective of consumer innovation regulation, for consumers with strong innovation, buying and consuming green food earlier and more frequently than others are important behaviors that showcase their individuality. Therefore, driven by this personality trait, the amplification of online green interactions, flow experiences, and self-efficacy stimulates their purchasing intentions. From the perspective of information trust regulation, when consumers have doubts about the authenticity of online green information, their acceptance of the information decreases, hindering the internalization of knowledge and experience in the information into self-efficacy. This, in turn, triggers consumer avoidance of purchases.

### 5.2. Marginal Contribution

The marginal contributions of this research lie in two aspects:

Firstly, compared to previous research paradigms that only consider rational cognition as the sole decision-making mechanism, this study incorporates emotional factors as a core category into the behavioral model. This study, based on a sample of Chinese food consumers, demonstrates to some extent the significant role of emotions in the context of food consumption in China. It offers valuable insights for distinguishing food consumption patterns across different countries and regions. Additionally, given the similar cultural and historical backgrounds shared by other Asia-Pacific countries such as Vietnam, Singapore, Japan, and South Korea, this study may serve as a useful reference for understanding food consumption behavior in these regions as well. Further, it comprehensively examines the dual mediating paths of “rationality-emotion” and explores the mediating mechanisms between consumers’ online green interaction and their willingness to purchase green food from the perspectives of emotional and cognitive dual influences, respectively, by investigating the mediating effects of flow experience and self-efficacy, two types of psychological variables. In addition, previous studies exploring the decision-making mechanism of rational cognition often explain green consumption behavior from the perspective of product cognition or environmental cognition. However, this research takes “environmental self-efficacy” as a research variable and explores the explanatory power of consumers’ self-perception on green food consumption.

Secondly, this study examines whether there is a connection between consumers’ rational cognition system and emotional intuition system, that is, whether the flow experience brought about by interaction will awaken individuals’ environmental self-efficacy, thereby further stimulating a chain-mediating mechanism of consumers’ willingness to purchase green food. It is of reference significance for exploring the antecedent factors of green food consumption in the context of the internet era and revealing the “black box” of consumer psychological mechanisms. Moreover, this study combines consumer characteristics and information trust to explore the moderating mechanisms of consumer innovativeness and information trust in the thought processes of emotion and rational cognition, thus identifying the boundary conditions of green food purchase intention. It not only enriches the research framework of consumer behavior but also innovatively integrates the research perspectives into a “cognition-emotion-consumer personality” framework.

### 5.3. Practical Insights

This study targets online green food consumers, analyzing how various psychological factors influence their willingness to purchase green food in the context of online green interactions. It also explores the underlying reasons behind Chinese consumers’ willingness to buy green food in the internet era. Based on the findings, the paper offers the following specific recommendations.

Firstly, it is crucial to encourage active consumer engagement with green information. Governments should collaborate with social media platforms to create online spaces dedicated to green food and sustainability discussions. Businesses should contribute by regularly sharing environmental content. Offering rewards such as discounts or product trials can effectively incentivize consumer participation, driving interaction and enhancing online marketing potential.

Secondly, the emotional aspect of consumer decision making should be prioritized. A positive online environment can significantly boost interaction. This can be achieved by improving website navigation, enhancing user interface design, and removing barriers to smooth interaction. These enhancements not only increase user satisfaction but also foster emotional connections, encouraging sustained consumer engagement. Additionally, nurturing influential figures to lead environmental campaigns can strengthen consumers’ sense of efficacy.

Thirdly, building trust in information is vital. In an era of widespread misinformation, businesses should avoid controversial topics and exaggerated marketing when promoting green products. Governments should regulate the accuracy of green product information and combat misleading claims, creating an environment where consumers can trust the information they receive.

Lastly, conduct user profiling to identify innovative user groups, such as those characterized by youth, high interactivity, and openness to new products. Companies can then use AI-driven recommendation algorithms to deliver targeted green content to these users’ social media feeds.

### 5.4. Research Limitations and Future Implications

There are several issues that require further investigation in this study, and we believe that future research can be conducted in the following three directions: (1) This study employed an empirical research method, which has its own limitations. In the future, it can be combined with a situational experimental approach for further investigation. (2) Although this study leveraged established theories, such as the SOR model, to explore the impact of online green interactions on consumers’ intention to purchase green food, it did not incorporate qualitative research methods to identify influencing factors. This omission may lead to a limited reflection of the theoretical model’s specificity concerning green food purchases. (3) When conducting a questionnaire survey, the measurement of green food consumption behavior in this study was based on self-report by the respondents, which may result in deviations between the survey data and actual behavior, which is a common limitation in the field of consumer behavior. In the future, cross validation can be conducted through neuroscience research methods such as skin conductance response, eye-tracking experiments, facial expression analysis, etc.

## Figures and Tables

**Figure 1 foods-13-03001-f001:**
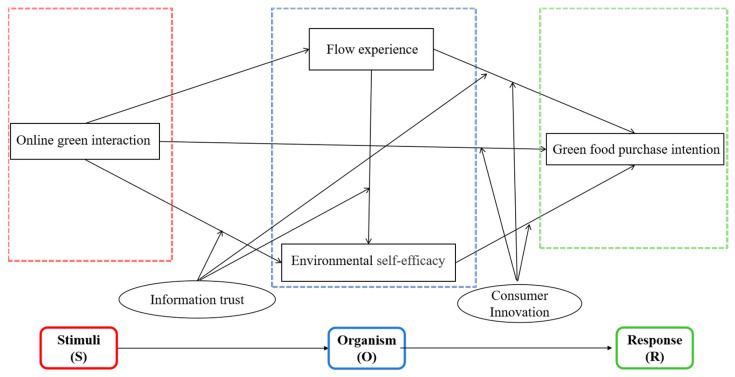
Research model framework diagram.

**Figure 2 foods-13-03001-f002:**
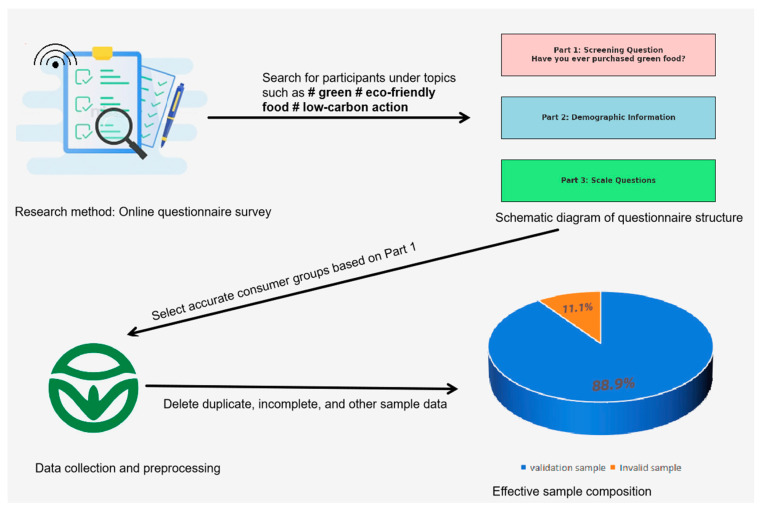
Research flowchart (#The theme of online topics).

**Figure 3 foods-13-03001-f003:**
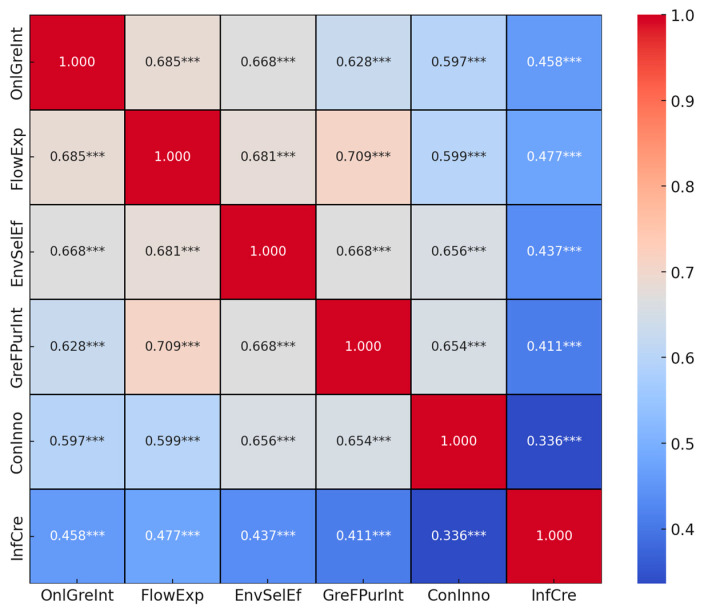
Correlation heatmap (*** *p* < 0.01).

**Table 1 foods-13-03001-t001:** Sample basic characteristics.

Variable	Specification	Sample (*n* = 407)	Percentage (%)
Gender	Female	242	59.5
	Male	165	40.5
Age (years)	19–30	222	54.5
	31–40	64	15.7
	41–50	63	15.5
	>50	58	14.3
Monthly Disposable Income (USD)	<420	187	45.9
	420–820	110	27.0
	820–1220	41	10.1
	1220–1620	29	7.1
	>1620	40	9.8
Education Level	Below Senior High School	14	3.4
	Senior High School or Technical Secondary School	30	7.4
	Undergraduate or Tertiary institutions	294	72.2
	Graduate Students or Above	69	17
Job	Civil Servants	11	2.7
	Enterprise Employees	90	22.1
	Public Institution Personnel	70	17.2
	Students	176	43.2
	Self-employed Employees	14	3.4
	Other Occupations (including Freelancers)	46	11.3

Notes: Tertiary institutions refer to adult professional and technical colleges established in China, with a general school system of 2–3 years, collectively referred to as higher education institutions with undergraduate universities.

**Table 2 foods-13-03001-t002:** Definition and explanation of variables.

Viable	Indicator	Code	Item	Source
Independent Variables	Online Green Interaction (OnlGreInt)	GI1	1. I am willing to engage with others on social media regarding green food.	Jiang [66] (2010), Nambisan [67] (2009)
		GI2	2. The information shared on social media has an impact on the purchase of green food.	
		GI3	3. I gather feedback on green food through environmental information on social media platforms.	
		GI4	4. I was attracted by the green food information shared on social media platforms.	
Dependent Variables	Green Food Purchase Intention (GreFPurInt)	PI1	1. I frequently purchase green food.	Kumar [68] (2015)
		PI2	2. I firmly believe in the performance of green food, and I am willing to purchase it even if its price is higher.	
		PI3	3. I would opt for purchasing environmentally friendly green foods.	
Mediating Variables	Flow Experience (FlowExp)	FE1	1. The information on green food available on social media platforms is of great interest to me.	Chan [69] (2013)
		FE2	2. When browsing or viewing information related to green food on social media, I feel that time passes quickly.	
		FE3	3. When browsing through information about green food on social media, I often pay particular attention to it.	
	Environmental Self-Efficacy (EnvSelEf)	SE1	1. I believe I have the capability to contribute to the attainment of environmental objectives.	Du [70] (2022), Chen [71] (2001)
		SE2	2. I believe I can effectively fulfill the environmental mission.	
		SE3	3. I believe I am capable of effectively addressing environmental issues.	
		SE4	4. I believe we can discover innovative approaches to address environmental issues.	
Regulatory Variables	Information Credit (InfCre)	IC1	1. The information on social media is accurate and factual.	Kim [72] (2021)
		IC2	2. The information on social media is reliable.	
		IC3	3. The information on social media is trustworthy.	
	Consumer Innovation (ConInn)	CI1	1. I am interested in experimenting with novel and innovative products and features.	Lao [53] (2013)
		CI2	2. I am interested in perusing diverse information and news regarding novel products.	
		CI3	3. I am interested in studying and understanding the changes and characteristics of new products.	

**Table 3 foods-13-03001-t003:** Reliability and convergent validity of variables.

Variable	Number	Mean	SD	Cronbach α	AVE	CR
OnlGreInt	4	2.862	0.546	0.834	0.666	0.888
FlowExp	3	3.775	0.811	0.904	0.839	0.940
GreFPurInt	3	3.942	0.691	0.812	0.728	0.889
EnvSelEf	4	3.807	0.713	0.894	0.760	0.927
InfCre	3	4.066	0.638	0.958	0.973	0.973
ConInn	3	3.097	0.948	0.880	0.806	0.923

**Table 4 foods-13-03001-t004:** Results of confirmatory factor analysis.

Model	Factors	χ^2^/df	RMSEA	GFI	CFI	IFI	TLI
Six Factors	OnlGreInt, EnvSelEf, InfCre, ConInn, FlowExp, GreFPurInt	2.321	0.057	0.912	0.969	0.969	0.962
Five Factors	OnlGreInt, EnvSelEf, InfCre, ConInn, (FlowExp + GreFPurInt)	3.138	0.072	0.876	0.948	0.948	0.938
Four Factors	OnlGreInt, EnvSelEf, InfCre, (FlowExp + GreFPurInt)	3.501	0.078	0.885	0.949	0.949	0.939
Three Factors	(OnlGreInt + EnvSelEf), (FlowExp + InfCre), (ConInn + GreFPurInt)	11.523	0.160	0.680	0.731	0.732	0.694
Two Factors	(OnlGreInt + FlowExp + EnvSelEf), (InfCre + ConInn + GreFPurInt)	13.301	0.173	0.544	0.682	0.683	0.643
One Factor	(OnlGreInt + EnvSelEf + InfCre + ConInn + FlowExp + GreFPurInt)	13.729	0.176	0.638	0.669	0.670	0.630

Note: The parentheses in the “factor” column indicate the integration of latent variables within them into a single one. The full name of GFI is goodness of fit index; CFI stands for Comparative Fit Index; RMSEA stands for Root Mean Square Error of Estimation; TLI stands for Test of Logical Interpretation, and IFI stands for Incremental Fit Index.

**Table 5 foods-13-03001-t005:** Mediation effect test (*n* = 407).

	(1)	(2)	(3)	(4)	(5)
Variables	GreFPurInt	GreFPurInt	GreFPurInt	GreFPurInt	GreFPurInt
EnvSelEf				0.466 ***	0.301 ***
				(0.048)	(0.050)
FlowExp			0.457 ***		0.346 ***
			(0.041)		(0.044)
OnlGreInt		0.812 ***	0.351 ***	0.419 ***	0.209 ***
		(0.51)	(0.061)	(0.062)	(0.063)
Age	0.064	0.054	0.0194	0.054 *	0.030
	(0.038)	(0.030)	(0.026)	(0.030)	(0.025)
Educ	0.041	0.001	−0.011	0.001	−0.027
	(0.065)	(0.051)	(0.043)	(0.051)	(0.043)
Gender	0.044	0.054	0.022	0.023	0.029
	(0.076)	(0.059)	(0.052)	(0.059)	(0.049)
MonSalary	0.049	0.008	0.004	0.008	0.003
	(0.033)	(0.026)	(0.023)	(0.026)	(0.022)
Constant	3.322 ***	1.273 ***	0.975 ***	1.273 ***	0.586 ***
	(0.266)	(0.245)	(0.217)	(0.245)	(0.217)
R-squared	0.024	0.402	0.736	0.634	0.762
F-value	2.521 *	53.885 ***	78.980 ***	53.885 ***	79.015 ***

Note: Standard errors in parentheses; *** *p* < 0.01, * *p* < 0.1.

**Table 6 foods-13-03001-t006:** Test of moderation effect (*n* = 407).

	(1)	(2)	(3)	(4)	(5)	(6)
Variables	GreFPurInt	GreFPurInt	GreFPurInt	GreFPurInt	EnvSelEf	EnvSelEf
InfCre×					0.110 ***	
OnlGreInt					(0.045)	
InfCre×				0.063 **		0.063 **
FlowExp				(0.030)		(0.030)
ConInn×			0.198 ***			
OnlGreInt			(0.060)			
ConInn×		0.263 ***				
EnvSelEf		(0.050)				
ConInn×	0.118 ***					
FlowExp	(0.042)					
InfCre				−0.187	−0.208	−0.143
				(0.127)	(0.137)	(0.126)
ConInn	−0.027	−0.623 ***	−0.068			
	(0.163)	(0.205)	(0.176)			
GrSelEf		−0.656 ***				
		(0.212)				
FlowExp	−0.076			0.398 ***		0.343 ***
	(0.183)			(0.095)		(0.094)
OnlGreInt			−0.355		0.415 ***	
			(0.255)		(0.145)	
Control	yes	yes	yes	yes	yes	yes
R-squared	0.772	0.756	0.734	0.718	0.691	0.703
F-value	84.208 ***	75.930 ***	66.621 ***	60.752 ***	52.137 ***	55.842 ***

Note: Standard errors in parentheses; *** *p* < 0.01, ** *p* < 0.05.

## Data Availability

The data presented in this study are available on request from the corresponding author.

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
