# Peer review of "Psychological Changes in Green Food Consumption in the Digital Context: Exploring the Role of Green Online Interactions from a Comprehensive Perspective"

_foods, 2024, doi:10.3390/foods13183001_

Round 1
Reviewer 1 Report
Comments and Suggestions for Authors
The whole introduction is very much contextualised to the experience and evidence of a well-defined part of the globe, China. It would be very important to also try to include concepts from the literature of other continents, precisely to detail the specificity of China in this context.
This is important also in the conclusion part.
Some minors:
Line 34: strong sentence about the realtion between green food and safety. Can we put some reference here?
Line 52: right, but try to explore also the concept of “Self Sphere”. this concept tries to show that yes behaviour is strongly influenced by the external environment but there is an internal part that determines free will.
Reviewer 2 Report
Comments and Suggestions for Authors
The manuscript is interesting, I would especially like to congratulate the very didactic figures: figure 1, 2 and figure 3 (and in #3, the manner of showing the strength of correlation with Heatmap was remarkable), but I have a few considerations:
1- I didn't find the description of the sample size calculation, I think it would be really important, as well as making sure that all the people were from the same region);
2- I noticed that people under the age of 18 responded, which is worrying. There are some social media that prevent this public from responding. How was the authorisation from the ethics committee? Did it allow this public to respond?
3- The SPSS version is 22.0. Please check, it seems to be very old. The current version is 29.0
4-In table 4, footnote the acronyms (e.g. GFI, CFI, etc.).
